# Participant Training for Virtual Reality User Studies: Lessons Learned and Open Questions

Jordan Allspaw
University of Massachusetts Lowell
Lowell, Massachusetts, USA
Jordan_Allspaw@uml.edu

Gregory LeMasurier
University of Massachusetts Lowell
Lowell, Massachusetts, USA
gregory_lemasurier@student.uml.edu

Elizabeth Phillips
George Mason University
Fairfax, Virginia, USA
ephill3@gmu.edu

Holly A. Yanco
University of Massachusetts Lowell
Lowell, Massachusetts, USA
holly@cs.uml.edu

## ABSTRACT

In the domain of human-robot interaction (HRI), teleoperating robots is a difficult task that often requires training to achieve proficiency. When Virtual Reality (VR) is used as a teleoperation method, there may be additional difficulties to achieving good performance due to general unfamiliarity with VR, and lack of standardized controls. With these factors in mind, properly training participants in studies utilizing VR is very important. In this paper, we will discuss VR teleoperation from a general research perspective, take inspiration from video games, and finally analyze training in our own user study. We conclude with recommendations for the community, as reporting our training methods and outcomes will allow us to develop best practices for VR teleoperation.

## CCS CONCEPTS

• **Human-centered computing** → **Virtual reality**; • **Computer systems organization** → **Robotic control**.

## KEYWORDS

Human-robot interaction (HRI), Training, Robot Teleoperation, Virtual Reality (VR)

**ACM Reference Format:**
Jordan Allspaw, Gregory LeMasurier, Elizabeth Phillips, and Holly A. Yanco. 2024. Participant Training for Virtual Reality User Studies: Lessons Learned and Open Questions. In *Proceedings of HRI-24 Workshop on Virtual, Augmented, and Mixed Reality for Human-Robot Interaction (Virtual, Augmented, and Mixed Reality for HRI).* ACM, New York, NY, USA, 7 pages. https://doi.org/10.1145/nnnnnnn.nnnnnnn

## 1 INTRODUCTION

Although advances within the robotics domain continue to enable robots to autonomously assist people with their lives both in the home and workspace, there are still many robot applications that

*Virtual, Augmented, and Mixed Reality for HRI, March 2024, Boulder, CO, USA*
© 2024 Association for Computing Machinery.
ACM ISBN 978-x-xxxx-xxxx-x/YY/MM...$15.00
https://doi.org/10.1145/nnnnnnn.nnnnnnn

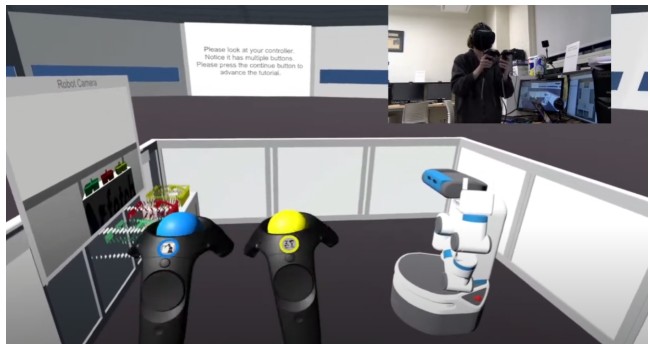

**Figure 1: In this screenshot from our study's VR training composited with an image of the person, the participant is starting the virtual reality training.**

remain impractical for robots to complete without human intervention. Tasks such as bomb defusal, search and rescue, or space exploration are difficult to automate due to the complexity of the task, risk of failure, and challenging environments. In such cases, robots are often teleoperated by skilled operators. A variety of teleoperation interfaces can be used by operators depending on the robot and task, ranging from traditional methods such as gamepads or keyboard and mouse, to newer methods such as Virtual Reality (VR).

Further, some VR interfaces have different interaction patterns for teleoperating the robot, including direct control [3, 4, 12, 23, 25] and waypoint-based interfaces [9, 15, 19, 26]. In direct control interfaces, operators move a robot in real time, whereas waypoint-based interfaces require an operator to set way-point goals and approve a robot's plan to reach each goal prior its execution.

Several groups have investigated the effectiveness of using VR for robot teleoperation. Direct control VR teleoperation interfaces have been found to result in higher usability [25], lower workload [25], and faster rate of operator's task completion [3, 25] when compared to a keyboard and mouse interfaces. Hetrick et al. [9] compared two VR teleoperation methods and found that waypoint-like controls were more beneficial to novice operators than direct control. Pryor et al. [19] proposed a waypoint-based VR interface and found that it reduced operator workload without any impact on performance compared to a keyboard and mouse interface. Finally,

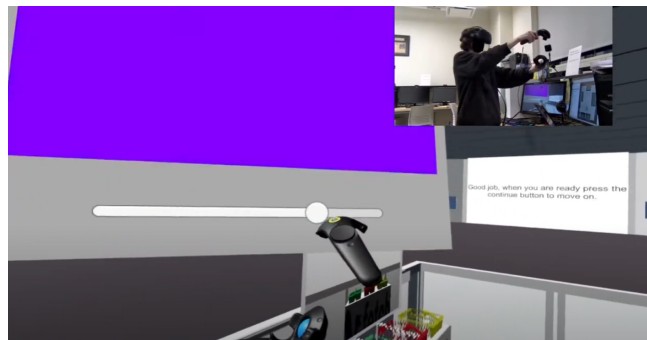

**Figure 2: In this screenshot from our study's VR training composited with an image of the person, the participant is completing the wristwatch slider tutorial, to learn how to interact with the wristwatch.**

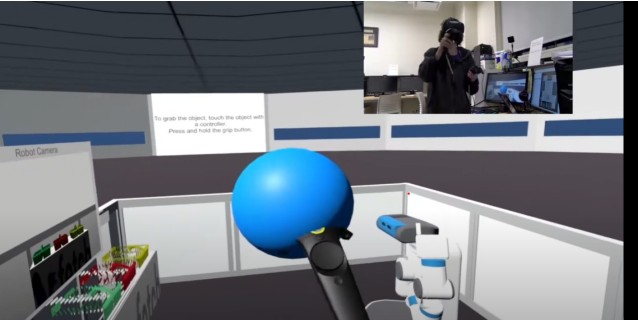

**Figure 3: In this screenshot from our study's VR training composited with an image of the person, the participant is completing the grasping and manipulating tutorial.**

we have previously proposed a user study to compare a VR and keyboard and mouse planner interfaces across difficult navigation and manipulation tasks [14], the results of which are currently awaiting review [16].

However, regardless of interface type, teleoperating robots is difficult, especially for novices [20]. While interfaces should be developed to be intuitive and require less training, some amount of training will always be necessary [20]. Even amongst experts, a lack of training can result in failures [10] and more training is correlated with better performance [18].

Training is especially important when introducing users to technology with which they may not be familiar. While consumer VR is becoming more available, the technology has not yet reached the availability of traditional interface methods, leading to the average person being less familiar with the technology. In addition, users that are familiar with VR may only be familiar with a single control scheme. When you combine a lack of familiarity with the technology as well as with the fact that controlling robots is already difficult, having an effective training protocol is necessary for users to be able to effectively teleoperate robots to perform tasks.

In our prior study [14, 16], participants followed a scripted training scenario[1] meant to familiarize them with the VR equipment, as well as aspects of the interface that they would later use to control the robot. The training session was unbounded, i.e., participants were not restricted by any time or performance constraints. Participants were able to ask specific questions of the experimenter while performing the training task, and were able to move through the different stages at their own pace. The overall goal of this training session was to allow each participant to feel reasonably proficient at the task before moving on to the experimental tasks. While conducting our study, we observed that people had varying performance during the training session. Thus, we were interested in further investigating effective training protocols for VR.

To explore this space, we examined other research with VR teleoperation experiments, how the video game industry is currently training new users in VR, and our own observations from our own

user study. The goal of this paper is to compile knowledge from these different sources in order to foster a discussion about designing effective training for robot teleoperation, as well as to encourage transparency and reporting of training procedures when describing findings of user studies that leverage these technologies.

In the following sections we present insights gained from reviewing the VAM-HRI literature, evaluating VR-based video games as an analogous source of training information, and present an analysis of results from a user study that explore the relationships between individual differences in training and performance.

## 2 TRAINING FOR VR TELEOPERATION IN HRI STUDIES

It is essential to consider training when developing an interface for robot teleoperation. Training procedures can potentially influence the ways people use interfaces and how effective they are at doing so. To investigate training procedures used within the VAM-HRI community, we conducted a literature review of papers published in VAM-HRI from 2018 to 2023.

First we included papers that were published at VAM-HRI, used VR for robot teleoperation, and mentioned or reported a user study. This resulted in 10 included papers. Of these papers, only six mentioned that training was part of their procedure, and, out of these six papers, only one provided information regarding training time and one reported enough information to replicate their training procedure.

Many VAM-HRI papers propose ongoing or late-breaking work, so we also expanded our search to the references in those papers and the papers that cited all of the VAM-HRI papers that used VR. Similar to our initial survey, only papers that used VR for robot teleoperation, and mentioned or reported a user study were included in this analysis. This resulted in 10 included papers. Of these papers, four mentioned that training was part of their procedure. Out of these four papers, none provided information regarding training time and two reported enough information to replicate their training procedure.

Of the surveyed papers, one common approach to providing instructions during training was to show information on world-anchored panels [12, 14]. In a paper outside of our surveyed set, experimenters have also used verbal instructions and descriptions

---

[1]https://www.youtube.com/watch?v=bnNPh5dkTbE&list=PLfUzSIwyYwvUw0YTkqgNDsSsq-P8Ts8sV&index=1

of the system's limitations for training [21]. As the way that information and instructions are communicated to a person during training can potentially influence engagement and understanding, it is important to report on this.

The tasks chosen for training can also potentially influence performance and understanding. Some groups train on a similar task to the one in the experimental runs [12, 14, 25]. One notable training task was using direct manipulation of a robot, in compliance mode, to understand the robot's capabilities [25]. While this enables the user to understand and explore limitations of the robot's capabilities, this approach is not feasible in every situation as the robot would need to be safe and compliant.

The elements of the interface that are covered in the tutorial can also potentially influence performance. Some groups give participants time to practice using the complete interface [1, 12, 14]. In some training sessions, participants are given time to learn equipment [1, 4, 14, 23].

While we may seem to be critical of the prior work in this area, we note that, even in our own work, we often gloss over training procedures due to limited page lengths. However, since training can influence people's use of and performance with an interface, it is important that we, as a community, are more transparent about these procedures. We recommend that researchers provide more information regarding their training sessions in appendices or through providing video links of the training procedure. (See Section 6 for additional details.)

In order to develop and understand what makes an effective training procedure, we conducted the above literature review wishing to identify how much time participants spent training, how instructions and information are conveyed to participants, what tasks participants completed in training, and what interface elements were taught as part of the training. If we, as a community, were to better document training procedures, we could better understand what comprises an effective training procedure, if any aspects of training can predict performance, and if we might derive best practices for VR training. We further discuss documented aspects of training procedures within the surveyed literature below.

## 3 LEARNING FROM GAMING

There are likely many lessons that can be learned from video games that might help to address problems that are unique to HRI. For instance, commercial-off-the-shelf games can introduce engagement with tasks, induce different cognitive states in players, and also create cognitive fidelity needed to transfer the skills developed in training to the real-world [17]. Thus, we investigated a variety of VR games, looking specifically at certain key factors of how they train the user. There are many different games available, so, to narrow our list, we utilized a list of popular VR games provided by IGN [5] which gave a total of 11 games of different genres. We excluded Tilt Brush from our analysis as it did not have a tutorial and is a 3D painting interface. The data from the other 10 games can be found in Table 1. This list contains a variety of VR focused games, as well as traditional games that have been reformatted to be played in VR. Analyzing these games is helpful because it shows what kind of prior experience users who are familiar with VR will

likely have. It also demonstrates the training methods that gaming companies currently believe are effective.

### 3.1 Controls

When looking at the video games from the list, each of them were analyzed for how they introduced the control systems and interactions to the player. In some cases, this was through a dedicated tutorial level. In other cases, it was simply the first part of game-play, where information is given to the player. Based on the games analyzed, none of them chose to display tutorial information permanently attached to the character's HUD, as is commonly done in traditional video games. Four of the games chose to create virtual windows with text or images/animations on them to present information to the users, while five chose to present text and indicators directly into the world, in an augmented reality style, with one game using both methods. In addition five of the games provided audio instructions to the user, either through a disembodied narrator, or through a virtual avatar. For the actual display, in all 11 cases there was text presented to the user, however in three cases there were images also presented, and in two cases there were also animations presented to the user, demonstrating actions.

### 3.2 Navigation

For navigation, two of the games allowed the user to teleport around the arena through a pointer, one allowed the user to travel using a joystick, similar to traditional videogames, three had the character not able to move large distances themselves, and instead the game itself moved the player to new locations periodically, with four games simply locking the player to one region with no major movement.

Looking at these games can provide some insight for what methods of information might be useful. In particular, showing persistent text on the HUD is probably not desirable, however depending on the situation both text/images on floating windows and text/images floating in AR can be valid, however just floating text on windows can be less engaging. Overwhelmingly top games are moving away from navigating with a joystick, similar to traditional video games, and are instead favoring locking the character in one location, that either doesn't move or is moved by the game in certain situations, such as game checkpoints. Even teleportation wasn't common, although was slightly favored compared to joystick movement.

## 4 ANALYSIS OF TRAINING IN OUR USER STUDY

In our prior discussion about our user study reported in [14], we explained that we wanted to explore best methods for human-in-the-loop command and control interface for a mobile manipulator robot. We wanted to compare the performance of VR against a traditional keyboard and mouse interface for teleoperating a robot, specifically in a complex environment. There have been other comparison studies for specific types of tasks such as manipulation [2, 22, 23, 25] and navigation [1], but our goal was to evaluate a single interface across both complex manipulation and navigation tasks.

While conducting the user study, we noticed that participants' training time varied greatly, where some participants seemed to struggle more than others. In this work, we extended our prior

**Table 1: Approaches to information presentation in during VR training in video games. The listed games are from a top VR games list from IGN [5].**

| Game | Text Attached to Floating Window | Text Attached to world AR | Audio from Narrator or Avatar |
|---|---|---|---|
| Beat Saber | Yes | No | No |
| Superhot | No | Yes | No |
| Space Pirate Trainer | Yes | No | No |
| Robo Recall | No | Yes | Yes |
| Job Simulator | No | No | Yes |
| Moss | No | No | Yes |
| Creed: Rise to Glory | Yes | Yes | Yes |
| Vader Immortal | No | Yes | Yes |
| Apex Construct | No | Yes | No |
| Journey of the Gods | Yes | No | No |

analysis [16], focusing just on the 23 participants who used the VR interface, in order to investigate factors that influence training time and how training time influences performance. We analyzed the amount of time spent during the unbounded training time, on learning the controls, the navigation planner, the manipulation planner, and their overall training time. We describe the metrics we captured related to training and performance below.

## 4.1 Variance in Training Time

Participants' overall training times ($M = 26.73, SD = 11.09$) ranged from 15.10 minutes to 58.05 minutes. This wide range demonstrates that while some participants completed our training session with ease, others struggled.

We originally expected that training would not take more than 20 minutes, based on observations from pilot testing, which included people with varying VR experience. Out of our 23 participants, only five completed the training within our expected time frame.

## 4.2 Prior Experience and Training Time

After observing the variance in training time, we anticipated that a participant's self reported prior experience may provide insights. Participants responded to a set of seven point Likert-type items, ranging from strongly disagree (1) to strongly agree (7) to evaluate their prior experience with robots, radio controlled vehicles, first person perspective video games, real time strategy games, PlayStation or Xbox controllers, virtual reality, and 3D modeling software. We ran Pearson's correlations between each of these experience measures and participants' training times, but we did not find any correlations. One potential reason may be that people with experience might be familiar with control schemes that were different than those used in the study, requiring them to learn the new control scheme.

## 4.3 Risk and Training Time

Participants also filled out the risk seeking sub-scale from Grasmick et al.'s [6] low self-control scale. The internal consistency of this sub-scale was Cronbach's $\alpha = 0.744$.

We found that participants' risk seeking scores were correlated to their navigation (Pearson's $r = 0.443, p = 0.044$) and trended to correlate with their manipulation (Pearson's $r = 0.411, p =$

0.072) training times. This shows that participants who seek out risk tended to take longer to complete the training for the robot's planners, potentially because they created riskier plans, resulting in failed plans.

There were no correlations between self-reported risk seeking and time spent learning the controls (Pearson's $r = 0.132, p = 0.591$) or to their total training time (Pearson's $r = 0.359, p = 0.143$).

Notably, the risk seeking scores did not correlate to performance metrics including points scored (Navigation Points: Pearson's $r = -0.009, p = 0.975$; Manipulation Points: Pearson's $r = -0.148, p = 0.585$) or number of collisions (Navigation Points: Pearson's $r = -0.318, p = 0.150$; Manipulation Points: Pearson's $r = 0.145, p = 0.530$) during the actual experimental tasks.

## 4.4 Spatial Orientation Skills and Training Time

Our VR interface required users to move around virtual goals to enable a robot to perform tasks in 3D space. These types of tasks would seemingly require spatial orientation skills. We measured participants' spatial orientation skills using the Perspective Taking/Spatial Orientation (PTSOT) [8, 11] mean error scores, which evaluates one's ability to imagine different perspectives from different locations in space.

We found correlations between PTSOT mean error scores and the total time spent on training (Pearson's $r = 0.521, p = 0.027$). These correlations persisted through the subsets of training, including controls (Pearson's $r = 0.497, p = 0.03$), navigation (Pearson's $r = 0.508, p = 0.019$), and manipulation (Pearson's $r = 0.479, p = 0.032$) times. These correlations suggest that people with lower spatial orientation skills (higher mean error scores) took longer to complete the training.

## 4.5 Performance and Training Time

Training time was correlated to participants' performance during the actual experimental task trials. Participants who took longer to complete the entire training scored fewer points overall (Navigation Points: Pearson's $r = -0.799, p = 0.001$; Manipulation Points: Pearson's $r = -0.569, p = 0.042$). This correlation holds for most subsets of training time, including training time spent on controls (Navigation Points: Pearson's $r = -0.805, p < 0.001$; Manipulation Points: Pearson's $r = -0.593, p = 0.033$), time spent learning the navigation

planner (Navigation Points: Pearson's $r = -0.763, p = 0.001$), and time spent learning the manipulation planner (Navigation Points: Pearson's $r = -0.639, p = 0.014$). Points scored during the manipulation task trended with time spent learning the navigation planner (Manipulation Points: Pearson's $r = -0.486, p = 0.078$), but were not correlated with time spent learning the manipulation planner (Manipulation Points: Pearson's $r = -0.434, p = 0.121$).

There was also a correlation between the number of collisions during the navigation task and total training time (Pearson's $r = -0.475, p = 0.046$) where longer training time correlated with fewer collisions. There were trending correlations between collisions during the navigation task and the subsets of training time, including training time spent on controls (Pearson's $r = -0.434, p = 0.072$), time spent learning the navigation planner (Pearson's $r = -0.421, p = 0.064$), and time spent learning the manipulation planner (Pearson's $r = -0.423, p = 0.063$). There were no correlations between the number of collisions during the manipulation task and time spent on training.

These correlations suggest that people who took longer during the training were likely slower during the actual tasks and thus scored fewer points and had fewer collisions.

## 4.6 Performance and Spatial Orientation Skills

With these observed correlations, one would believe that since lower spatial and orientation skills result in longer training time, and since longer training time trends to result in people having fewer collisions and points scored in navigation tasks and fewer points in manipulation tasks, then lower spatial and orientation skills may trend with fewer collisions and points scored across tasks. We ran a correlation test for users' spatial and orientation skills compared to the number of collisions they experienced during the navigation (Pearson's $r = -0.329, p = 0.134$) and manipulation (Pearson's $r = -0.168, p = 0.466$) tasks. We also ran a correlation test for users' spatial and orientation skills compared to the number of points they scored during the navigation (Pearson's $r = -0.393, p = 0.132$) and manipulation (Pearson's $r = -0.435, p = 0.092$) tasks. There were no observed correlations.

## 4.7 Workload and Training Time

In our study, participants filled out the NASA-TLX [7] to measure their subjective cognitive workload after completing all navigation and all manipulation trials. There was a correlation between the total time participants spent training and their workload after the navigation task (Pearson's $r = 0.658, p = 0.01$). This holds for the subsets of training time, including training time spent on controls (Pearson's $r = 0.68, p = 0.007$), time spent learning the navigation planner (Pearson's $r = 0.711, p = 0.002$), and time spent learning the manipulation planner (Pearson's $r = 0.659, p = 0.005$). This shows that people who took longer in the training had a higher experienced workload after the navigation task. There was not a correlation between training time and the workload experienced after the manipulation task.

## 5 OPEN QUESTIONS

Through our investigation on training within the VR teleoperation space, we have identified several open questions that we should work towards addressing as a community.

### Q1: How should control information be conveyed to an operator?

As seen in our review, there are several different ways to provide information to an operator during training. This includes panels with text, audio from a narrator or virtual avatar, or verbal tutorials from an experimenter. The way information is conveyed to an operator may influence their engagement and understanding of the covered material, therefore further investigations into these modalities are necessary.

### Q2: What are effective tasks for training?

Designing tasks on which to train users is a difficult problem. Tasks that are too similar to the experimental tasks can influence learning effects, whereas knowledge gained from tasks that are too different from the experimental task may not translate as well. These tasks should also enable participants to explore the range of the robot and the interface's capabilities, so that they understand the functionality ahead of time and are not learning new concepts during experimental runs.

### Q3: How do we know when a person has been properly trained and can proceed with the study?

Often, user studies that focus on VR teleoperation of robots are investigating performance-based metrics across interfaces or control methods. Training can impact a person's performance with an interface, therefore it is essential for all users to be properly trained on the interface before the experimental runs. People learn at different rates; if participants are not given sufficient time to learn during training, then some participants will still be learning the concepts – that others learned during training – in their experimental runs, potentially influencing results. It is also important to consider the trade-off between training time, fatigue, workload, and performance. People who take longer to train may enter the experimental runs with more fatigue than those who quickly learned the interface, which may in turn reduce their performance.

### Q4: Should training be constant or adapt to the person's needs?

With constant training, everyone gets the same experience and is taught the same concepts. However, everyone enters training with different prior experiences and knowledge which may help them better or more easily understand some concepts covered in training compared to someone without those experiences. Alternatively training can be dynamic and adaptive. For example, when users make errors, an interface can provide more details to explain why the error occurred and how to prevent similar errors in the future [13, 20, 24]. With this approach, users who successfully completed a component of training without fully understanding the underlying concepts being taught may enter the experimental runs with less knowledge. The trade-off between constant training,

where user's prior knowledge and experience may have a larger influence on results, and adaptive training, where not everyone goes through the same training experience potentially influencing results, should be explored further.

## 6 RECOMMENDATIONS

In order to determine effective methods for training users to tele-operate robots in VR and to learn how various training approaches can influence performance, we encourage the community to report more information regarding their training procedure and metrics. While this information is often excluded due to page length limits, we encourage authors to add a supplementary link to this information as a footnote in their paper to enable future evaluations, comparisons, and for best practice standards to be developed for training procedures.

Based on our analysis of research within the VAM-HRI research community, as well as complementary information from video games and our own user study, we believe that it could be useful to include how long the participant training lasts (expected time as well as the minimum, maximum, mean, and standard deviation for participants). Furthermore, documenting how information was presented to the participant (e.g., verbally or through a virtual scenario). Describing whether participants received robot- or task-specific training as opposed to training on general VR controls also would provide insight into how training for the study was conducted.

By provided this information on training, future researchers could compare success across different training approaches in different user studies, as well as provide documentation for future studies on best practices.

## 7 CONCLUSION

In this paper we sought to open a discussion about how to design effective VR training for a teleoperated robot user study. We looked at literature from other VR interfaces discussed at VAM-HRI, took inspiration from video games, and presented results from our own user study. We also believe that there are some potentially interesting user studies that could be conducted to investigate the effects of training – both the type of training and its length – on performance after training.

Due to the correlations between training time and risk seeking and spatial orientation skills, and the correlations between training time and performance data, there is some ability to predict when participants will likely take longer to complete tasks prior to their actual attempts. Partially due to the general public's unfamiliarity with VR, we feel that there are still many open questions to investigate to determine what constitutes an effective and efficient training system for VR. We hope this paper will foster discussion in our community and be useful to others looking to run user studies. In addition we would also encourage those in the VAM-HRI community to release information about training when conducting a user study, to foster effective training.

## ACKNOWLEDGMENTS

The training time analysis uses data from the VR portion of a prior interface comparison study [16]. Thank you to our collaborators, James Tukpah, Murphy Wonsick, Brendan Hertel, Jacob Epstein, S. Reza Ahmadzadeh, and Taskin Padir for their contributions on the user study. Thank you to Nam Bui and Danny Machado for assisting with data collection.

This work was supported in part by the National Science Foundation (IIS-1944584) and the Office of Naval Research (N00014-21-1-2582 and N00014-18-1-2503). The views, opinions, and/or findings expressed are those of the authors and should not be interpreted as representing the official views or policies of the Department of Defense or the U.S. Government.

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
