# OpenReview forum: "Participant Training for Virtual Reality User Studies: Lessons Learned and Open Questions"
_humanrobotinteraction.org/HRI/2024/Workshop/VAM-HRI — VAM-HRI 2024 Oral_

### Official Review · Reviewer_k1fF · 2024-02-24
**Accept**

**Rating:** 8
**Confidence:** 5

**Review:**

The paper discusses the importance of effective training in Virtual Reality (VR) for teleoperating robots in human-robot interaction studies. It highlights the challenges due to unfamiliarity with VR and the lack of standardized controls by doing a detailed review of existing VR teleoperation training in HRI and gaming. A previous user study was conducted by the authors to compare VR and keyboard and mouse planner interfaces across difficult navigation and manipulation tasks. A subset of the results obtained from this for the training time in VR was used to provide a detailed analysis in this paper. This analysis explores factors like training time variance, the impact of prior experience, risk, and spatial orientation skills, and the correlation between training time and performance. It concludes with recommendations for reporting training methods and outcomes to develop best practices for VR teleoperation.

Strengths:

1. Comprehensive Analysis: The paper provides an in-depth analysis of training time and its impact on performance in VR teleoperation, contributing valuable insights to the field.
2. Practical Recommendations: It offers practical recommendations for the community to improve transparency and reporting of training procedures.


Areas of Improvement:

1. The findings are based on a specific user study, which may limit their applicability to other contexts or setups. Details regarding the user study were also missing from the paper, thus, making it difficult to form a background idea of the involved experiment.
2. Further empirical studies could strengthen the recommendations and findings, ensuring they are robust across different VR teleoperation contexts.
3. A small typo - training has been mentioned twice in paragraph 3 of page 2.


In summary, I think this paper is a good fit for VAM-HRI, and I recommend acceptance.

---

### Official Review · Reviewer_srv3 · 2024-02-26
**Accept**

**Rating:** 7
**Confidence:** 5

**Review:**

This paper presents an overview of pre-interaction participant training techniques for VR within experimental contexts. It presents a mixture of findings from a prior VR teleoperation user study by the authors, along with summary analysis of training elements in a collection of similar VR teleoperation papers, and popular VR video games. The paper concludes with a series of open questions regarding user training for VAM-HRI contexts, and calls for improved reporting of training details in experiments for the field at-large.

Strengths:
- This paper touches on an underexplored aspect of VAM-HRI (as evidenced by the lack of reproducible documentation of training procedures throughout papers in the field), which will hopefully spur a more nuanced discussion of this design problem going forward.
- A discussion of best practices for VR training is relevant, not only for HRI experiments, but also for VR systems intended for long-term deployment.
- The synthesis of the varied analyses into open questions is a helpful format for readers considering the design of their own training procedures.

Weaknesses:
- There's a bit of a disconnect in paper scope, as the only VAM use-cases under investigation (from the authors' prior work and the survey of VAM literature) were specifically VR for robot teleoperation. This conflicts with both the title, which mentions the broader "Participant Training for Virtual Reality User Studies," as well as the collected data from the 10 popular VR video games, none of which include teleoperating a robot as a gameplay element. Broadening the scope of HRI papers examined by including analysis of training within the wider world of VAM papers that include a user study, but involve any kind of human-robot interaction (not only teleoperation), would rectify this mismatch and improve the analysis in the paper.
- The analysis of training methods in video games could go into further depth: as of now, the primary data concerns details about how textual information is presented (in a floating window, attached to world AR, or with audio), and how the player is able to move. These largely only correspond to Q1 of the Open Questions section - Q2-4 also have equivalents in video game design, which would make for interesting discussion.

This paper is very relevant to VAM-HRI and will generate a good deal of discussion about an aspect of VR study design that requires further attention, due to the large effect it can have on experimental results. Hopefully this paper will have its intended effect, and lead to improved reporting about training for studies in VAM-HRI. I recommend this paper for acceptance and presentation at this year's VAM-HRI workshop.

---

### Decision · Program_Chairs · 2024-02-26

Accept (Oral)